# The Tumor Necrosis Factor-α Level in Platelet-Rich Plasma Might Be Associated with Treatment Outcome in Patients with Interstitial Cystitis/Bladder Pain Syndrome or Recurrent Urinary Tract Infection

**DOI:** 10.3390/ijms25010163

**Published:** 2023-12-21

**Authors:** Jia-Fong Jhang, Yuan-Hong Jiang, Teng-Yi Lin, Hann-Chorng Kuo

**Affiliations:** 1Department of Urology, Hualien Tzu Chi Hospital, Buddhist Tzu Chi Medical Foundation and Tzu Chi University, Hualien 970, Taiwan; alur1984@hotmail.com (J.-F.J.); redeemerhd@gmail.com (Y.-H.J.); 2Department of Laboratory Medicine, Hualien Tzu Chi Hospital and Buddhist Tzu Chi Medical Foundation, Hualien 970, Taiwan

**Keywords:** biomarkers, prognosis, proteins, inflammation

## Abstract

Using platelet-rich plasma (PRP) injections to treat urological diseases has attracted great attention. This study investigated the impact of cytokine concentrations in PRP on the treatment outcome of patients with recurrent urinary tract infection (rUTI) and interstitial cystitis/bladder pain syndrome (IC/BPS). Forty patients with IC/BPS and twenty-one patients with rUTI were enrolled for four-monthly repeated PRP injections. PRP was collected at the first injection and analyzed with multiplex immunoassays for 12 target cytokines. In patients with IC/BPS, a Global Response Assessment (GRA) score ≥ 2 was defined as a successful outcome. In rUTI patients, ≤2 episodes of UTI recurrence during one year of follow-up was considered a successful outcome. Nineteen (47.5%) patients with IC/BPS and eleven (52.4%) patients with rUTI had successful outcomes. The IC/BPS patients with successful outcomes had significantly lower levels of tumor necrosis factor-α (TNF-α) in their PRP than those with unsuccessful outcomes (*p* = 0.041). The rUTI patients with successful outcomes also had a lower level of TNF-α (*p* = 0.025) and a higher level of epidermal growth factor (*p* = 0.035) and transforming growth factor-β2 (*p* = 0.024) in PRP than those with unsuccessful outcomes. A lower level of TNF-α in PRP might be a potentially predictive factor of treatment outcome.

## 1. Introduction

Platelet-rich plasma (PRP) is an autologous blood product, which is defined as a portion of plasma with a platelet concentration above the baseline [1]. Since the 1970s, PRP has been gradually applied in maxillofacial surgery and in treating musculoskeletal and dermatological diseases. This is because of its various potential biological properties, such as tissue adherence, homeostatic action, anti-fibrosis, anti-inflammation, and promotion of tissue regeneration [1,2]. Although the molecular mechanism of PRP in the abovementioned biological properties remains unclear, recent studies have revealed that satellite cells, as well as myogenic and non-myogenic interstitial cells, may represent a direct target for PRP action [3]. Several cytokines and growth factors, including platelet-derived growth factor (PDGF), transforming growth factor (TGF)-β, and epidermal growth factor (EGF), have been identified as key factors in mediating the tissue regeneration mechanism of PRP [3,4]. In a previous clinical study, the level of TGF-β in PRP correlated with symptomatic and radiological improvement in tendon healing in patients who received PRP injections [5]. The concentration of growth factors and cytokines in PRP might affect treatment efficacy, and is potentially a predictive factor of treatment outcome in clinical practice. However, studies to investigate the association between cytokine concentration in PRP and treatment outcome are still limited. PRP has been widely used to treat various diseases, and the key factors in PRP that have an important impact on therapeutic outcome should be different between different diseases.

Interstitial cystitis/bladder pain syndrome (IC/BPS) is a chronic disease that is characterized by bladder pain and urinary frequency without evidence of a urinary tract bacterial infection [6]. The histological characteristics of bladders in patients with IC/BPS revealed urothelium denudation and inflammation [7]. Recurrent urinary tract infection (rUTI) is a common urological problem, which highly negatively impacts a patient’s quality of life. Previous studies have revealed persistent bladder inflammation and unhealed urothelium in rUTI patients who have been asymptomatic after antibiotic treatment [8]. Both IC/BPS and rUTI are challenging urological problems, as disease relapse is common after conventional treatment [6,8]. We previously conducted two clinical trials using intravesical PRP injection to treat patients with IC/BPS and rUTI [9,10]. Improvement in clinical symptoms and laboratory urothelial health were noted in rUTI and IC/BPS patients who were refractory to conventional treatment, and the overall treatment success rate was approximately 50–60% in both studies [9,10]. Although clinical studies reporting the efficacy of PRP in treating urological disorders are still limited, many urologists in Taiwan and China have used PRP to treat their patients as an off-label use. The preliminary results of using PRP to treat patients with rUTI and IC/BPS are promising, but the predictive factors for determining treatment outcome are unknown. Selecting appropriate patients to receive PRP treatment is an important issue in clinical practice, and cytokine concentration in PRP might have important impact on treatment outcomes. This study aims to investigate the cytokine and growth factor concentrations in the PRP of IC/BPS and rUTI patients, and to analyze their impact on intravesical PRP treatment outcome.

## 2. Results

A total of 40 IC/BPS and 21 rUTI patients underwent four-monthly intravesical PRP injections, and all of the patients completed the follow-up course. The mean ages of the IC/BPS and rUTI patients were 51.6 ± 15.3 and 70.6 ± 7.6 years old, respectively. According to the abovementioned definitions, 19 of the 40 patients with IC/BPS (47.5%) and 11 of the 21 patients with rUTI (55%) had successful treatment outcomes. No PRP injection was associated with adverse effects, such as acute cystitis, severe bladder pain, or hematuria after the procedure. 

The mean ICPI, ICSI, and VAS in patients with IC/BPS was significantly improved after four repeated PRP treatments (ICPI: 9.3 ± 2.8 to 7.1 ± 2.3, *p* = 0.045; ICSI: 9.4 ± 3.5 to 7.1 ± 2.5, *p* = 0.036; VAS: 4.2 ± 2.3 to 2.6 ± 1.7, *p* = 0.013). In patients with IC/BPS, the overall GRA was 1.4 ± 0.8. The age and platelet count in the PRP were not significantly different between the IC/BPS patients with successful and unsuccessful outcomes (Table 1). The baseline clinical characteristics, including cystometric bladder capacity, VAS pain score, maximal bladder capacity during cystoscopic hydrodistention, and the grade of bladder glomerulation were also similar between the IC/BPS patients with successful and unsuccessful outcomes. The platelet count in peripheral whole blood was higher in the IC/BPS patients with unsuccessful outcomes, but the ratio of platelet count in PRP to whole blood was not different between the two groups.

Among the 21 patients with rUTI, the mean number of UTI episodes in the 12 months before PRP treatment was 5.8 ± 2.9, and the mean number of UTI episodes in the 12 months after PRP treatment was 4.19 ± 2.02 (*p* = 0.016). Eleven of the rUTI patients (52.4%) had successful outcomes (UTI episodes ≤ 2) in one year after the fourth PRP injection. The mean GRA was 1.05 ± 1.13. The rUTI patients with successful outcomes were younger and had fewer UTI episodes in the previous one year than those with unsuccessful outcomes (Table 1). The platelet count in whole blood and PRP was similar between the rUTI patients with successful and unsuccessful treatment outcomes. The ratio was also similar between the two groups. 

Table 2 shows the targeted cytokine levels in PRP in patients with IC/BPS with successful and unsuccessful treatment outcomes. Among the 12 targeted cytokine and growth factors in PRP, only the level of TNF-α was significantly lower in IC/BPS patients with successful outcomes (*p* = 0.041). Table 3 shows the targeted levels of cytokine in the PRP of patients with rUTI. The level of TNF-α in PRP was also significantly lower in rUTI patients with successful outcomes (*p* = 0.025), while the levels of EGF, PDGF-AA, and TGF-β2 in PRP were significantly higher in rUTI patients with successful outcomes (*p* = 0.035, 0.041, and 0.024, respectively). 

Age was not significantly correlated with any cytokine concentration in both patients with IC/BPS or rUTI. In the patients with IC/BPS, the platelet count was significantly correlated with the level of IL-8, PDGFAB/BB, and TNF-α in PRP (*p* = 0.001, r = 0.537, and *p* = 0.031, r = 0.351; *p* = 0.032, r = 0.348, respectively, based on Pearson correlation regression analysis). In the patients with rUTI, the platelet count was not significantly correlated with any cytokine concentration in the PRP. Table 4 shows the significant correlations between PRP cytokine levels and symptom changes in patients with IC/BPS and rUTI. Among the cytokines, the level of IL-4 was positively correlated with the change in VAS, ICPI, and ICSI. The GRA in IC/BPS patients was significantly negatively correlated with TNF-α (*p* = 0.001, R = −0.628) and IFN-α2 (*p* = 0.03, R = −0.444). In rUTI patients, the levels of PDGF-AA (*p* = 0.023, R = 0.517), PDGFAB/BB (*p* = 0.032, R = 0.494), and TGF-β1 (*p* = 0.02, R = 0.528) were significantly positively correlated with GRA, while TNF-α was negatively correlated with GRA (*p* = 0.04, R = −0.421). The number of UTI episodes in one year after PRP treatment was negatively correlated with TGF-β2. 

ROC curves were constructed for the abilities of PRP cytokines to distinguish among treatment outcomes. The TNF-α level in PRP provided an acceptable value for discriminating a successful treatment outcome in rUTI patients (cut-off value: 14.54, sensitivity = 0.889, specificity = 0.727, AUC = 0.798, Figure 1A). For IC/BPS patients, the TNF-α level in PRP only provided a marginal discriminating ability for a successful outcome (cut-off value = 4.82, sensitivity = 0.636, specificity = 0.643, AUC = 0.633, Figure 1B). 

## 3. Discussion

Recently, we published several preliminary studies to show promising results in using PRP to treat IC/BPS and rUTI patients who were refractory to conventional treatment; however, only about 50–70% patients had a successful treatment outcome [9,11]. Since intravesical PRP injection is a relatively invasive treatment, selecting suitable patients to receive PRP treatment is an important issue. Theoretically, the level of cytokines in PRP should have impact on the treatment outcome, but this association has not been investigated. The current study revealed that both rUTI and IC/BPS patients with unsuccessful treatment outcomes had higher TNF-α levels in their PRP than patients with successful outcomes. The level of TNF-α was negatively correlated to patients’ satisfaction with PRP. The TNF-α level in PRP is a potential predictive factor regarding treatment outcome when using PRP to treat patients with unhealthy urothelium and bladder inflammation.

Early studies suggest that higher platelet concentrations in PRP was associated with better effects when using PRP for bone regeneration [12]. However, recent studies revealed conflicting results [13]. Although the efficacy of PRP treatment should be credited to the cytokines and growth factors contained in the PRP [2,4], clinical studies investigating the association between growth factor concentration and treatment outcome are rare. In clinical practice, PRP is usually made by various commercialized kits, but most kits do not provide information on the concentration of platelet and growth factors in the PRP. Previously, limited evidence revealed that PDGF-AB and TGF-β were associated with increased symptom improvement in the knees of patients with osteoarthritis who underwent PRP therapy [5,14]. In patients who underwent PRP treatment for alopecia, the glial cell line-derived neurotrophic factor concentration in PRP was positively correlated with hair density [15]. The age of patients might also be a factor associated with PRP treatment outcome. Taniguchi et al. reported that age was negatively correlated with the PDGF and insulin-like growth factor-1 concentration in PRP, but the impact on treatment outcome remains unknown [16]. The clinicians who use PRP injection to treat patients should pay attention to the quality of PRP. 

Although the pathogenesis of IC/BPS is unclear, previous studies with various laboratory evidence revealed that urothelium denudation and inflammation are common characteristics in IC/BPS bladders [6,17]. Evidence also revealed that unhealthy urothelium is an important factor for rUTI recurrence [8]; however, treatment for promoting urothelium recovery from bacterial infection is also lacking. PRP injection has been widely used in regeneration medicine for its ability to eliminate chronic inflammation and promote cell proliferation [1,18]. However, many growth factors in the PRP might contribute to the therapeutic effect; which factor is the most important key factor in treating functional bladder diseases is still unknown. 

TNF-α is a powerful pro-inflammatory agent and plays a pivotal role in orchestrating the cytokine cascades in many inflammatory diseases [19,20,21]. An interesting study showed that TNF-α can stimulate mouse intestine epithelial cell proliferation at physiological concentrations, but inhibits proliferation at higher pathological concentrations [22]. Anti-TNF-α agents have recently been used to treat patients with IC/BPS. A recent randomized, placebo-controlled study proved its clinical efficacy regarding improving bladder pain and urinary frequency [23]. Since TNF-α might aggravate inflammation and inhibit epithelial proliferation, higher levels of TNF-α in PRP might impede its efficacy in treating patients with inflammatory bladder diseases such as IC/BPS and rUTI. Recently, several preclinical studies found downregulation of the TNF-α level in tissues that undergo PRP treatment [24,25], but the role of PRP TNF-α concentration in the therapeutic effect has not been investigated. Our current study firstly demonstrated that the level of TNF-α in PRP was associated with treatment outcome and a lower TNF-α level might be a predictive factor of better therapeutic efficacy in using PRP to treat inflammatory bladder diseases.

Since PRP has various biological properties for treating many diseases with different pathogeneses, the functional content of PRP should be different among the different diseases. Although our results revealed that lower TNF-α levels in PRP are associated with better outcomes for PRP treatment of IC/BPS and rUTI, this result may not be applicable to other diseases. This study also found that several factors of PRP are also associated with clinical outcomes and correlate with clinical symptoms, such as IL-4 level for patients with IC/BPS, and TGF-β2 for patients with rUTI. Although there is a lack of consistency in the two patient groups, these factors potentially affect the PRP treatment results. Some of the cytokines in the PRP may have a specific effect on bladder symptoms, for example, a higher level of IL-4 in PRP may have the effect of better bladder pain improvement.

The main limitations of this study are the small case number and lack of a placebo control group. The cytokine concentration in whole blood was not investigated. The difference in cytokine level in PRP may be a direct result of its level in whole blood. A recent study revealed that immune cell profiles in peripheral blood and serum cytokine levels in patients with IC/BPS were associated with disease severity [26]. Our patients received four-monthly repeated PRP injections, but the cytokine levels were only investigated at the first PRP administration. The consistency of cytokine levels in serial PRP treatment remains unknown. The patients with IC/BPS did not stop their use of oral medications for their bladder symptoms, which also might have impact on the outcome of PRP treatment. We only enrolled 12 kinds of cytokine for investigation of their concentration in PRP, and some cytokines we have not investigated yet may also impact therapeutic outcomes. 

## 4. Material and Methods

### 4.1. Patients

This study enrolled patients with IC/BPS and rUTI who underwent intravesical PRP injection from 2017 to 2020. Patients were considered to have rUTI if they experienced ≥2 bacterial cystitis episodes in the past 6 months or ≥3 infections within the preceding 12 months [27]. The rUTI patients suffered from the disease for more than 1 year and had previously received prophylactic managements such as life style modification, increased water intake, long-term antibiotics (more than 3 months) or non-antibiotic prophylaxis. However, the patients still suffered from frequently recurring UTIs. Before the study enrollment, all of the rUTI patients had received a full course of antibiotic treatment to treat their acute cystitis. Urine analysis and urine culture were obtained from the patients with rUTI, and only the patients without pyuria or bacteriuria could be enrolled into this study. The exclusion criteria for the rUTI patients in this study included spinal neurogenic voiding dysfunction, bladder outlet obstruction, vesicoureteral reflux, and a history of using immunosuppressants. 

The IC/BPS patients were diagnosed with the AUA guideline for IC/BPS: “An unpleasant sensation (pain, pressure, discomfort) perceived to be related to the urinary bladder, associated with lower urinary tract symptoms of more than six weeks duration, in the absence of infection or other identifiable causes” [6]. The IC/BPS patients had previously received treatment including cystoscopic hydrodistention, oral pain control medication, and intravesical hyaluronic acid installation. All patients underwent complete medical history and clinical symptoms investigation before PRP treatment. The exclusion criteria for the IC/BPS patients included recent acute cystitis, detrusor overactivity, neurogenic voiding dysfunction history of using immunosuppressant, and bladder outlet obstruction. The IC/BPS patients with Hunner’s lesions were also excluded. The study was performed in one center and was approved by the ethics committee of the hospital (Institutional Review Board number: TCGH 105-48-A and 106-173-A). All of the patients were informed of the potential benefits and risks of PRP treatment and signed the informed consent before enrollment. All parts of this study comply with the Declaration of Helsinki.

The IC/BPS patients were investigated using the visual analog scale (VAS) for pain and the O’Leary-Sant symptom score including symptom (ICSI) and problem indices (ICPI) at baseline, then at three months after the fourth PRP injection. The Global Response Assessment (GRA) was used to evaluate treatment outcomes in both IC/BPS and rUTI patients. It contains a seven-point symmetric scale: markedly worse (−3), moderately worse (−2), slightly worse (−1), no change (0), slightly improved (+1), moderately improved (+2), and markedly improved (+3). In IC/BPS patients, GRA ≥ 2 at three months after the fourth PRP injection was defined as a successful treatment outcome [9]. The rUTI patients were followed for one year after the fourth PRP injection, and ≤2 episodes of UTI recurrence during the 1-year follow-up was considered a successful treatment outcome [10]. 

### 4.2. PRP Preparation and Injection Procedures

Both rUTI and IC/BPS patients underwent four-monthly intravesical injections of 10 mL PRP at 20 sites under general anesthesia in the operation room. The PRP preparation procedure was previously reported in our studies [9,10]. Briefly, 50 mL whole blood was withdrawn from the patients and was sent to our central laboratory, where the technologist (TY Lin) centrifuged the whole blood with a first soft spin (190× *g*, 20 min, <20 °C). Then, the supernatant plasma containing platelets was transferred into another sterile tube without disturbing the buffy coat. The platelet-containing plasma was centrifuged again by a hard spin (2000× *g*, 20 min, <20 °C). The platelet pellets were formed at the bottom of the tube; the lower third was PRP and the upper two thirds were platelet-poor plasma. Platelet-poor plasma was removed, and the platelet pellets were added to the plasma by shaking the tube to form 10 mL sterile PRP. The PRP injection procedure also had been previously reported [9,10]. Both IC/BPS and rUTI patients received 20 intravesical suburothelial injections of the PRP solution. The volume of each PRP injection site was 0.5 mL PRP. The injection depth was about 1 mm in the suburothelium, using a 23 gauge needle and rigid cystoscopic injection instrument (22 Fr, Richard Wolf, Knittlingen, Germany). 

### 4.3. Evaluation Cytokines Concentration in the PRP

Before the first PRP preparation, 2 mL whole blood was analyzed for the total platelet count in our clinical laboratory department. After the first PRP preparation was completed, 1 mL PRP was collected from all patients to analyze the total platelet count and cytokine concentration. The collected PRP for cytokine concentration was stored at −80 °C. It was investigated with commercially available microspheres with the MILLIPLEX Human cytokine/chemokine magnetic bead-based panel kit (Millipore, Darmstadt, Germany). A total of 12 targeted growth factors and cytokines were analyzed with the multiplex kit, including EGF, PDGF-AA, PDGFAB/BB, TGF-β1–3, interferon alpha 2 (IFNα2), tumor necrosis factor-α (TNF-α), interleukine-4 (IL-4), IL-8, IL-13, and IL-17A. The quantification of targeted analytes in PRP samples was performed in accordance with the manufacturer’s instructions and the detailed procedures had been reported in our previous study [11]. Briefly, 25 μL assay buffer, 25 μL PRP sample, and 25 μL beads were added into 96-well plates (panel kits) and were incubated overnight at 4 °C. The contents in the wells then were removed, followed by washing twice with 200 μL wash buffer in the kit. An amount of 25 μL of detection antibody of each target was added into the wells, and the plates were incubated in the dark on a shaker plate for 1 h. An amount of 25 μL of streptavidin-phycoerythrin solution was added into each well; incubation was then performed in the dark for 30 min. The contents in the plates were removed again, followed by washing two times with 200 μL wash buffer. Finally, 150 μL of sheath fluid was added into the plates, and the plates were evaluated using the MAGPIX instrument with xPONENT version 3.1 software. Median fluorescence intensities of all targets were analyzed to calculate the corresponding concentrations in the PRP samples. All the laboratory procedures were approved by the Department of Medical Research of Hualien Tzu Chi Hospital, and were performed in accordance with relevant guidelines and regulations. 

The scheme of this study is showed in Figure 2. 

### 4.4. Statistics

Continuous variables are represented as means ± standard deviations. IC/BPS and rUTI patients were classified into successful and unsuccessful groups according to the abovementioned definition. The UTI episodes in the rUTI patients and bladder symptoms in the IC/BPS patients before and after treatment were compared with a paired *t*-test. Differences in clinical data and cytokine levels in PRP were analyzed using an independent *t*-test. Cytokines with mean values below the minimum detectable concentrations as per the assay manufacturer were excluded for further analysis. Receiver operating characteristic (ROC) curves were generated for each cytokine in order to distinguish patients with successful outcomes from those with unsuccessful outcomes. The areas under the ROC curves (AUC) were calculated. Linear Pearson correlation analysis was used to evaluate the correlation between the change in symptoms and the level of cytokines in the PRP. All statistical calculations were performed using SPSS Statistics for Windows, Version 20.0 (IBM Corp., Armonk, NY, USA).

## 5. Conclusions

Approximately 50% of rUTI and IC/BPS patients had successful treatment outcomes after intravesical PRP injections. Among the 12 cytokines and growth factors investigated in this study, the rUTI and IC/BPS patients with successful treatment outcomes had lower TNF-α levels in their PRP. The level of TNF-α in PRP might be a factor that affects therapeutic outcome in patients with inflammatory bladder diseases.

## Figures and Tables

**Figure 1 ijms-25-00163-f001:**
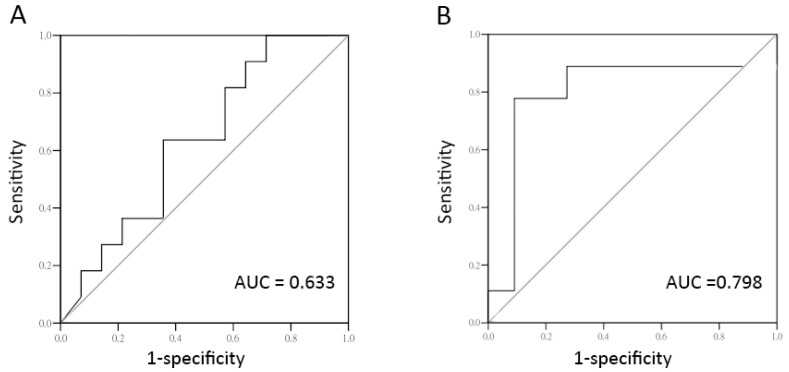
The level of TNF-α in PRP could be used to predict treatment outcome in patients with IC/BPS or UTI. (**A**) The ROC curve for using TNF-α level in PRP to predict successful outcomes in IC/BPS patients who underwent PRP injections. AUC = 0.633, cut-off value = 4.82, sensitivity = 0.636, and specificity = 0.643. (**B**) The ROC curve for using TNF-α level in PRP to predict successful outcome in rUTI patients who underwent PRP injections. AUC = 0.798, cut-off value = 14.54, sensitivity = 0.889, and specificity = 0.727.

**Figure 2 ijms-25-00163-f002:**
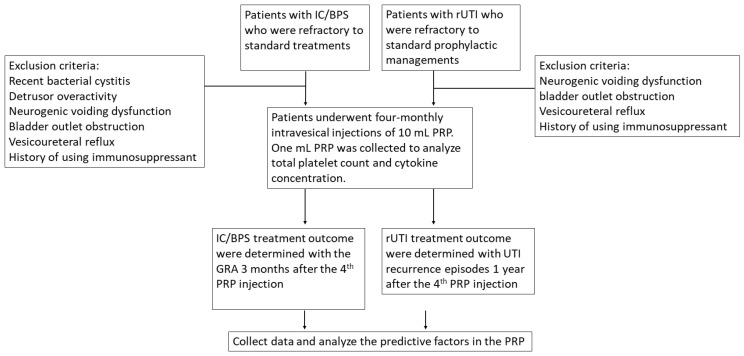
The scheme of study process, inclusion and exclusion criteria.

**Table 1 ijms-25-00163-t001:** Clinical parameters in the IC/BPS and rUTI patients who underwent intravesical PRP injection.

	**Overall IC/BPS** ***n* = 40**	**Successful IC/BPS** ***n* = 19**	**Unsuccessful IC/BPS** ***n* = 21**	***p*-Value**
Age (years old)	51.6 ± 15.3	51.37 ± 10.90	51.71 ± 18.65	0.940
ΔVAS	−1.6 ± 3.1	−1.5 ± 2.8	−1.7 ± 3.3	0.851
ΔICPI	−2.2 ± 5.2	−2.2 ± 5.7	−2.20 ± 5.05	0.992
ΔICSI	−2.4 ± 5.3	−1.6 ± 4.9	−2.9 ± 5.7	0.558
GRA	1.40 ± 0.82	2.18 ± 0.40	0.79 ± 0.43	<0.001
Platelet count in PRP (×10^3^/μL)	827.2 ± 308.9	767.3 ± 317.0	881.1 ± 299.1	0.264
Platelet count in whole blood (×10^3^/μL)	237.6 ± 76.3	206.1 ± 78.2	265.8 ± 64.0	0.014
Ratio	3.5 ± 1.1	3.6 ± 1.1	3.3 ± 1.0	0.416
	**Overall rUTI** ***n* = 21**	**Successful rUTI** ***n* = 11**	**Unsuccessful rUTI** ***n* = 10**	***p*-Value**
Age (years old)	70.6 ± 7.6	68.73 ± 3.8	72.7 ± 10.15	0.020
UTI episode in one year follow up	4.19 ± 2.02	3.09 ± 1.87	5.40 ± 1.43	0.005
GRA	1.05 ± 1.13	1.45 ± 0.93	0.50 ± 1.20	0.067
Platelet count in PRP (×10^3^/μL)	798.5 ± 311.2	690 ± 228.14	917.8 ± 356.67	0.159
Platelet count in whole blood (×10^3^/μL)	248.7 ± 83.0	223.91 ± 66.92	276 ± 93.57	0.359
Ratio	3.34 ± 1.10	3.26 ± 1.03	3.43 ± 1.23	0.778

ΔVAS: change in visual analog scale pain scale from baseline; ΔICPI: change in O’Leary-Sant score problem indexes from baseline; ΔICSI: change in O’Leary-Sant score symptom indexes from baseline; GRA: Global Response Assessment.

**Table 2 ijms-25-00163-t002:** The cytokines and growth factors concentration in the PRP in the IC/BPS patients with successful and unsuccessful outcome.

	IC/BPS Patients with Successful Outcome *n* = 19	IC/BPS Patients with Unsuccessful Outcome *n* = 21	*p*-Value
EGF	433.42 ± 377.01	441.80 ± 536.70	0.954
IFNα2	4.38 ± 3.69	5.57 ± 4.03	0.356
IL-4	2.65 ± 5.28	1.86 ± 1.87	0.545
IL-8	4.76 ± 3.10	4.37 ± 2.29	0.655
IL-13	26.40 ± 47.57	12.17 ± 7.94	0.214
IL-17A	2.84 ± 8.86	1.74 ± 5.18	0.641
PDGF-AA	6508.60 ± 6669.45	5265.69 ± 7239.42	0.580
PDGFAB/BB	36,695.37 ± 8848.76	38,200.00 ± 11,276.33	0.640
TNF-α	6.65 ± 4.91	10.15 ± 5.96	0.049
TGF-β1	76,573.11 ± 44,179.35	60,774.67 ± 29,569.14	0.198
TGF-β2	2795.62 ± 1784.72	2175.30 ± 1280.73	0.220
TGF-β3	54.29 ± 15.04	49.38 ± 14.69	0.303

**Table 3 ijms-25-00163-t003:** The cytokine and growth factor concentration in the PRP of rUTI patients with successful and unsuccessful outcomes.

	rUTI Patients with Successful Outcome *n* = 11	rUTI Patients with Unsuccessful Outcome *n* = 10	*p*-Value
EGF	433.29 ± 218.64	258.45 ± 220.3	0.035
IFNα2	7.15 ± 10.19	6.02 ± 6.75	0.526
IL-4	4.62 ± 4.2	2.97 ± 2.09	0.503
IL-8	8.79 ± 3.67	6.03 ± 3.09	0.091
IL-13	30.03 ± 20.54	24.87 ± 16.34	0.359
IL-17A	1.78 ± 3.08	1.21 ± 2.55	0.526
PDGF-AA	14,714.91 ± 4994.03	10,553.6 ± 7427.21	0.041
PDGFAB/BB	44,584.27 ± 9688.49	42,900.6 ± 24,971.27	0.084
TNF-α	61.49 ± 171.03	410.28 ± 1168.8	0.025
TGF-β1	116,744.27 ± 32,798.92	89,235 ± 28,607.66	0.072
TGF-β2	5232.55 ± 1636.82	3838.1 ± 1047.17	0.024
TGF-β3	68.37 ± 4.85	68.72 ± 5.39	0.853

EGF: epidermal growth factor; IFNα2: interferon alpha 2; IL: interleukin; PDGF: platelet-derived growth factor; TNF-α: tumor necrosis factor-α; TGF-β: transforming growth factor-β.

**Table 4 ijms-25-00163-t004:** The correlation between the cytokines concentration in PRP and the clinical parameters in the patients with IC/BPS and rUTI.

		**Patients with IC/BPS *n* = 40**											
		**EGF**	**IFNα2**	**IL-4**	**IL-8**	**IL-13**	**IL-17A**	**PDGF-AA**	**PDGFAB/BB**	**TNF-α**	**TGF-β1**	**TGF-β2**	**TGF-β3**
ΔVAS	R	−0.070	0.154	0.555	0.142	−0.015	0.436	−0.090	−0.042	0.043	0.012	−0.082	−0.276
	*p*-value	0.739	0.473	0.004	0.499	0.942	0.029	0.674	0.840	0.840	0.953	0.698	0.182
ΔICPI	R	−0.155	0.262	0.593	0.237	0.009	0.392	−0.177	−0.011	0.253	−0.159	−0.274	−0.321
	*p*-value	0.459	0.216	0.002	0.255	0.965	0.053	0.407	0.957	0.222	0.449	0.186	0.118
ΔICSI	R	−0.282	0.075	0.510	0.202	0.026	0.272	−0.148	−0.164	0.174	−0.157	−0.226	−0.244
	*p*-value	0.172	0.726	0.009	0.333	0.900	0.188	0.489	0.434	0.404	0.455	0.278	0.240
GRA	R	−0.099	−0.444	0.060	−0.001	0.199	0.092	0.212	−0.314	−0.628	0.232	0.154	0.306
	*p*-value	0.638	0.030	0.774	0.997	0.341	0.663	0.320	0.126	0.001	0.265	0.463	0.137
		**Patients with rUTI *n* = 21**											
		**EGF**	**IFN** **α2**	**IL-4**	**IL-8**	**IL-13**	**IL-17A**	**PDGF-AA**	**PDGFAB/BB**	**TNF-α**	**TGF-β1**	**TGF-β2**	**TGF-β3**
UTI episodes	R	−0.274	0.107	−0.295	−0.373	−0.184	0.079	−0.298	−0.007	0.204	−0.354	−0.493	−0.008
	*p*-value	0.229	0.643	0.194	0.096	0.424	0.733	0.190	0.974	0.388	0.115	0.023	0.972
GRA	R	0.446	−0.361	0.161	0.064	0.038	−0.336	0.517	0.494	−0.421	0.528	0.386	0.172
	*p*-value	0.056	0.129	0.510	0.796	0.876	0.159	0.023	0.032	0.040	0.020	0.103	0.482

EGF: epidermal growth factor; IFNα2: interferon alpha 2; IL: interleukin; PDGF: platelet-derived growth factor; TNF-α: tumor necrosis factor-α; TGF-β: transforming growth factor-β; ΔVAS: change in visual analog scale pain scale from baseline; ΔICPI: change in O’Leary-Sant score problem indexes from baseline; ΔICSI: change in O’Leary-Sant score symptom indexes from baseline; GRA: Global Response Assessment.

## Data Availability

Data are available by contact with the corresponding author.

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
