# Peer review of "The Tumor Necrosis Factor-α Level in Platelet-Rich Plasma Might Be Associated with Treatment Outcome in Patients with Interstitial Cystitis/Bladder Pain Syndrome or Recurrent Urinary Tract Infection"

_ijms, 2023, doi:10.3390/ijms25010163_

Round 1
Reviewer 1 Report
Comments and Suggestions for Authors
Authors should be congratulated for their work. The topic is interesting and intriguing.
The manuscript is well-written and easily readable. Minor grammar editing is required. The need for outcome biomarkers in such complicated conditions such as IC/BPS is an interesting topic.
i recommend to shorten a bit the title, cause it is too long.
Tables and figures are good
A minor revision is required
Comments on the Quality of English LanguageEnglish is fine. Only minor corrections are required
Author Response
Authors should be congratulated for their work. The topic is interesting and intriguing.
The manuscript is well-written and easily readable. Minor grammar editing is required. The need for outcome biomarkers in such complicated conditions such as IC/BPS is an interesting topic.
i recommend to shorten a bit the title, cause it is too long.
Tables and figures are good
A minor revision is required
Response: Thanks for your kind comment. We had revised the title of this manuscript: “The Tumor Necrosis Factor-α Level in Platelet-Rich Plasma Might be Associated with Treatment Outcome in the Patients with Interstitial Cystitis/Bladder Pain Syndrome or Recurrent Urinary Tract Infection”
Reviewer 2 Report
Comments and Suggestions for Authors
One of the major concern of this manuscript is the lack of a placebo group.
There also some minor aspects that may be improved.
1. The authors should add a scheme for inclusion and exclusion criteria.
2. The Materials and Methods section might be organized into subchapters.
3. What statistic program was used? The authors should specify
4. Did the authors respected the Declaration of Helsinki? It should be stated.
5. The Discussion section includes too many general data in the first half. It should be more focused on comparisons with other studies.
Comments on the Quality of English Language
Minor English issues
Author Response
One of the major concern of this manuscript is the lack of a placebo group.
Response: Thanks for your comment. Indeed, lack of placebo control is the main limitation of this study. We have mentioned this limitation in the discussion section. (page 10, line 327 to 328)
There also some minor aspects that may be improved.
- The authors should add a scheme for inclusion and exclusion criteria.
Response: Thanks for your comment. We had added the new figure 1 to illustrate the scheme of this study including the inclusion and exclusion criteria. (age 4, line 158 to 161)
- The Materials and Methods section might be organized into subchapters.
Response: Thanks for your comments. We had added some subheadings in the Materials and Methods section. (page 2, line 76; page 3, 116; page 3, line 133; page 4, line 163)
- What statistic program was used? The authors should specify
Response: Thanks for your question and sorry for missing the information. All statistical calculations were performed using SPSS Statistics for Windows, Version 20.0 (IBM Corp., Armonk, NY, USA). We had added above statement in the Statistics section. (page 4, line 174 to 176)
- Did the authors respected the Declaration of Helsinki? It should be stated.
Response: Thanks for your question. All parts of this study comply with the Declaration of Helsinki. (page 3, line 103 to 104)
- The Discussion section includes too many general data in the first half. It should be more focused on comparisons with other studies.
Response: Thanks for your comment. Currently, studies to investigate the role of cytokine concentration in the treatment outcome of PRP are still rare. We had deleted some general information in the Discussion section. (page 9, line 259 to 261; page 9, line 272 to 275; page 10, line 298 to 300)
Round 2
Reviewer 1 Report
Comments and Suggestions for Authors
Authors should be congratulated for their work. The topic is interesting and intriguing. They improved the quality of the manuscript according to reviewer suggestions. The paper is suitable for publication in its current form